Subject Areas:
engineering geology/petrology

Keywords:
alternating triaxial loading–unloading test, conventional triaxial loading–unloading test, fitting curve, stress–strain curve, destruction rule

Authors for correspondence:
Changbao Jiang
e-mail: jcb@cqu.edu.cn
Minghui Li
e-mail: mhli@cqu.edu.cn

# A new loading–unloading experimental method for simulating the change of mining-induced stress

Changbao Jiang[1,2], Xiaodong Liu[1], Minghui Li[1],
Huan Yu[3] and Minke Duan[1]

[1]State Key Laboratory of Coal Mine Disaster Dynamics and Control, Chongqing University, Chongqing 400030, People's Republic of China
[2]School of Earth and Environmental Sciences, The University of Queensland, St Lucia, Queensland 4072, Australia
[3]Chongqing SIIE Product Testing Co. Ltd, Chongqing, People's Republic of China

CJ, 0000-0003-0520-9754

Based on a self-developed triaxial seepage device, a new loading and unloading experimental method is proposed in this paper to eliminate sample variations. The results show that the strength of sandstone sample and the axial strain at failure increased with the increasing initial hydrostatic stress, but decreased with the increasing loading–unloading rate. Following the alternating loading–unloading test, the stress–strain curves of specimens advanced in wave form, the waves' volatility decreased and then increased. It is found that near the ultimate strength, volatility is the biggest, and the stability of waves' volatility increased along with the increasing initial hydrostatic stress. The similarity of stress–strain curves between the conventional loading–unloading tests and the alternating loading–unloading tests increased along with the increasing initial hydrostatic stress and the increasing initial velocity for the alternating loading–unloading method. Along with the increasing initial hydrostatic stress, the failure behaviour of the sandstone samples tested under loading–unloading methods changed from a tensile state to a tensile-shear coexisting state, and finally to a fully shear failure state. The degree of failure modes increased with the increasing loading–unloading rate.

## 1. Introduction

Deep rock mass is under a triaxial stress equilibrium state before an excavation. Excavating an opening disturbs the original *in situ* stress field, causing stress redistribution around the excavation.

As the excavation face progresses, the radial stress of the surrounding rocks decreases constantly; however, the tangential stress may increase, decrease or remain unchanged. By contrast, when the tangential stress increases and the radial stress decreases, the stress difference of rock mass would most easily reach its limit state at failure. Many studies have shown that the change in mining-induced stress caused by excavation can be simulated by various loading–unloading tests. At present, some achievements have been accomplished in the change of rock mechanical properties caused by various loading–unloading tests. Hua & You [1] found one of the basic methods to prevent rock burst is to release the strain energy in rock before excavation. Based on the conventional triaxial compression test and the unloading confining stress test, mechanical properties of rock in the condition of unloading were studied [2–5]. Chen et al. [6] analysed the dilatancy behaviours of salt rock through the unloading confining stress test, conventional uniaxial test and the triaxial compression test under the same deviatoric stress. The volumetric deformation of salt rock under unloading was greater than that under triaxial loading and less than that under uniaxial loading. Furthermore, based on the unloading theory and experimental data, they proposed a constitutive model of salt rock damage. To simulate the behaviour of rock deformation and its failure characteristics under loading and unloading conditions, the triaxial unloading tests on granite, migmatitic granite and limestone were designed by Huang et al. [7], finding that rock bursts during tunnelling in the high in situ stress area could be controlled or reduced by lowering the excavation rate or applying precautionary measures in order to control the displacement of surrounding rocks. Huang & Li [8] found the failure modes of rock and strain energy conversions under different conditions by similar triaxial unloading tests. Based on the triaxial unloading tests under different stress paths, the space–time evolution laws of acoustic emission and energy-releasing characteristics of rock during deformation were studied [9]. The permeability evolution of coal was researched [10–13], and the mechanical properties of coal were analysed [14–16]. To reveal the variation laws of physical mechanical parameters of rock, the experiments in different stress paths (loading axial stress and unloading confining stress, unloading axial stress and unloading confining stress) have been designed by Xie & He [17], finding that in the condition of unloading stress, the phenomenon of rock failure behaves more fiercely than that in the condition of loading stress. The tests in similar stress paths were carried out by Huang & Huang [18] on the man-made rock specimens with single or double cracks to study the characteristics and mechanisms of rock crack evolution during the underground excavation. They found that the strength of rock and the evolution of cracks are clearly influenced by both the inclination angle of individual cracks and the combination geometry of cracks. The tests in three different unloading paths (constant axial stress and unloading confining stress, loading axial stress and unloading confining stress, unloading axial stress and unloading confining stress) have been designed by Zhao et al. [19] to investigate the evolution characteristics and the conversion rates of strain energy. Dai et al. [20] also designed a similar experiment to investigate the mechanical characteristics of rocks. Furthermore, an experimental investigation on the strength, deformation and failure behaviour of coarse marble under six different paths was studied by Yang et al. [21]. Wong et al. [22] studied the inelastic behaviour and failure modes of six different sandstones. At low confining pressures, shear-induced expansion and brittle fracture were overserved. At higher confining pressures, shear enhanced compaction and fragmentation flow were observed. Experimental data show that the particle breakage pressure (which decreases with increasing porosity and particle size) provides a quantitative measure of brittleness. Wang et al. [23] carried out triaxial compression experiments and confining pressure unloading experiments. The results show that the unloading damage is more fragile, and the unloading damage is more serious to the sandstone. Zhu et al. [24] used the sandstone of Yangcheng Coal Mine as the research object, and used the MTS device to carry out the uniaxial loading and unloading experiment to analyse the evolution characteristics of the stress, strain, peak stress and macroscopic failure characteristics of the sandstone during the uniaxial cyclic loading and unloading test. Zhang et al. [25] studied the sandstone of Shamushu Coal Mine in Yibin City, Sichuan Province, and carried out triaxial compression experiments on sandstone samples with different loading and unloading speeds under different confining pressures. The strength, deformation and permeability characteristics of the samples were analysed. As the loading and unloading speed of the confining pressure increases, the centre of the Moiré stress circle of the sample moves to the right, and the ultimate strength, peak strain and residual stress of the sample gradually increase. Ding et al. [26] studied sandstones in Xin'an Coal Mine, Zaozhuang City, Shandong Province, and studied sandstones at different temperatures (200°C, 400°C, 600°C and 800°C) and different confining pressures (20, 30 and 40 MPa). The influence of mechanical properties on the rock damage evolution after high-temperature treatment under

unloading conditions was analysed. Yang *et al.* [27] used the red sandstone of Shandong Province as the research object. The MTS815 test system was used to carry out conventional triaxial compression and unloading confining pressure experiments on sandstone samples. The strength characteristics, failure characteristics and internal crack evolution process of sandstone were analysed. Baud *et al.* [28] studied the sandstones in Berea, Boise, Darley Dale and Gosford, and analysed the relationship between water and sandstone strength, and the failure mode of high pressure. Lu & Liu [29] studied the rock mass expansion characteristics of sandstone specimens by triaxial loading and unloading experiments. The sandstone of Gubei Coal Mine was used as the research object. Combined with physical experiments and theoretical analysis, it is observed that rock mass expansion can be divided into two categories: (i) pre-peak expansion caused by crack propagation, and (ii) peak expansion after rotation or separation of rock mass sliding. Then the peak expansion deformation is the main reason for the volume expansion of the rock. The confining pressure has an inhibitory effect on the expansion deformation. Liu [30] took the sandstone of Changhua Coal Mine in Zhangzhou City, Shanxi Province as the research object, and carried out triaxial compression experiments on sandstone samples with different confining pressures (4, 8 and 12 MPa) and different hydraulic pressures (0, 3, 6 and 9 MPa). An equation was fitted for the coupling effect of sandstone confining pressure and water pressure. It is found that the strength of fine sandstone rock increases with the increase of confining pressure, and the strength and cohesion of rock mass and the hydraulic pressure have a negative exponential relationship. Taheri *et al.* [31] conducted a three-axis monotonic and cyclic compression study on sandstone samples by means of a closed-loop servo-controlled testing machine, taking the sandstone in Hawkesbury, Canada as the research object. Peak strength changes, axial loads, constrained pressures, and axial and lateral deformations were analysed. However, these loading–unloading tests above simulating the change of mining-induced stress were carried out on different specimens, which cannot avoid the discreteness on them. There are two experimental methods used to study the loading rate dependency in the mechanical properties on a single specimen. The first method is increasing stepwise strain rate, it allows researchers to find the loading rate dependency from a single specimen, but choosing the time to increase strain rate requires complicated technology and experience [32]. The other one is the method of alternating strain rate between high and low rates [33,34], it has been applied to various rock types, providing a large amount of data [35–37]. Although alternating tests were carried on a single rock specimen, these are uniaxial experiments, which only represent the increase or decrease of vertical stress, so it is different from the actual change of mining-induced stress. Recently, a new compression test combining multi-stage confining pressure and alternating loading rate has been investigated [38], although this new test considers the confining pressure. The changes in mining-induced stress simulated by this new test are also different from the actual change of mining-induced stress. *In situ* excavation can be simulated by the loading–unloading test with loading axial stress and unloading confining stress at the same time. Therefore, in the premise of fixed ratio 1 : 10 (rate of loading axial stress/rate of unloading confining stress), the experiment was carried out on sandstone specimens by using alternating loading–unloading test under different initial hydrostatic stresses and different loading–unloading rates. Through the experiment, we compared the similarity between the fitting curves (by fitting the stress–strain curves obtained from alternating loading–unloading test) and the conventional stress–strain curves under different loading–unloading conditions. In the test, the conditions that the fitting curves could realistically simulate the conventional loading–unloading curves and different stress–strain curves of loading–unloading rate also could be obtained from a single specimen. Furthermore, the effect of the loading–unloading rate on the mechanical properties in the case of eliminating the dispersion of different specimens could been analysed. The change of mining stress caused by rock excavation can be simulated by different loading and unloading experiments [14]. Therefore, the mechanical properties of the different loading and unloading rate experiments were studied to provide a basic database for efficient and safe rock excavation.

# 2. Experiment set-up

## 2.1. Experiment apparatus

The experiment was implemented by using the self-made 'triaxial servo-controlled seepage equipment for thermo-hydro-mechanical coupling of coal containing methane' [39], as shown in figure 1.

royalsocietypublishing.org/journal/rsos　R. Soc. open sci. 6: 181964

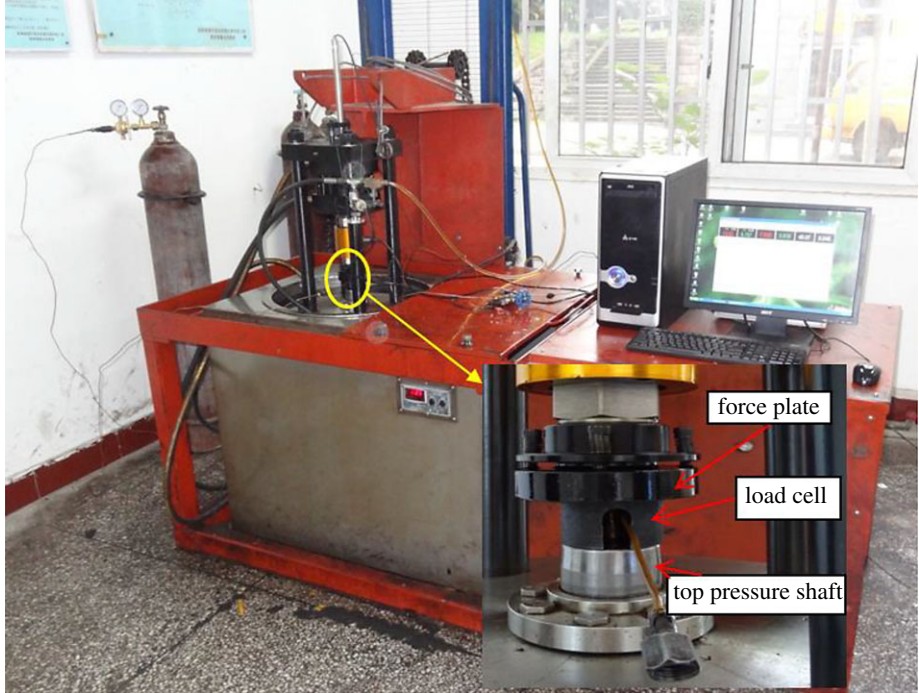

**Figure 1.** Triaxial servo-controlled seepage equipment for thermo-hydro-mechanical coupling of coal containing methane.

The maximum axial pressure of the device can reach 100 MPa. The maximum confining pressure can reach 10 MPa. The maximum gas pressure can be up to 2 MPa and the test temperature can be up to 90°C. Axial loads enable force control and displacement control. Parameters such as stress, deformation, gas pressure, temperature and flow can be collected automatically. The device can be applied to coal and other rock experiments, and can carry out mechanical and seepage experiments under different *in situ* stress fields (confining pressure and axial pressure), different pressures and different geothermal fields. Therefore, the sandstone samples were tested using this device.

## 2.2. Specimen preparation

The samples of this test are cylindrical specimens of the sandstone rock mass that were drilled by a drill with an inner diameter of 50 mm along a direction perpendicular to the rock bedding. The specimens were shaped into cylindrical specimens of $\Phi 50$ mm $\times$ 100 mm, without visible fractures and cracks based on the photo observation. The machining accuracy of the specimens was in accordance with the International Society for Rock Mechanics (ISRM) recommended methods [40]. The allowable variations of the end flatness and the deviation from perpendicularity to the longitudinal axis of the specimens were less than 0.02 mm and 0.001 rad, respectively. The sandstone samples of this experiment were taken from the roof of a coal mine in Chongqing, China. The apparent density ranged from 2237 to 2243 kg m$^{-3}$. The mineral composition mainly includes feldspar and quartz. The content of quartz is less than 75%, and the content of feldspar is more than 25%, and the amount of cuttings is less than 25% [41]. The sandstone has a moisture content of 0.44%, an ash content of 96.21%, a volatile content of 3.26%, a fixed carbon of 0.09%, a TOC content of 0.1% and a porosity of 4.15%. The sandstone has good homogeneity, small dispersion and good sampling integrity. The experimental samples were taken from the same intact large sandstone. Figure 2 is the photo of specimens.

## 2.3. Testing scheme

The specimens were divided into two schemes under different testing conditions, as listed in table 1. In the table, $\Delta\sigma$ represents the transforming gap of velocity. For example, the experimental scheme of sample 2-6-1 in table 1 shows that when the axial pressure reaches 6 MPa of initial hydrostatic pressure, the loading and unloading speed is set to 0.01 MPa s$^{-1}$, when the axial pressure reaches 9 MPa, the loading and unloading speed is converted to 0.1 MPa s$^{-1}$, and when the axial pressure

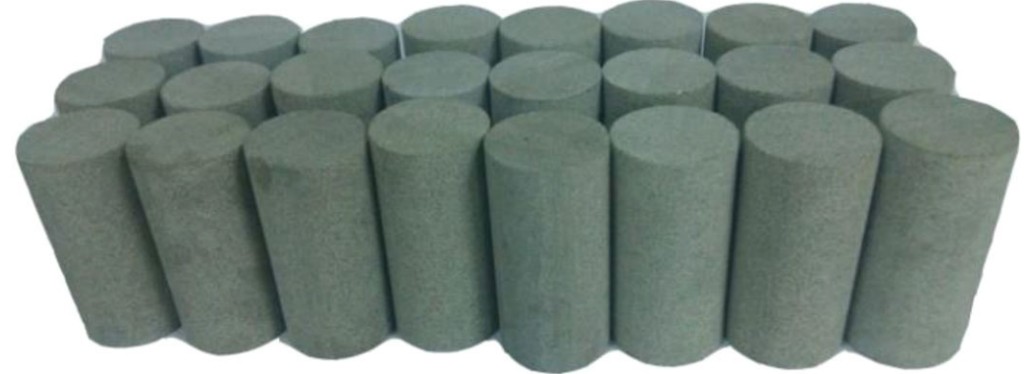

**Figure 2.** The specimens of sandstone.

**Table 1.** Parameters of triaxial loading – unloading test and mechanical properties of specimens.

| scheme | sample no. | $\Delta\sigma$ (MPa) | $\sigma_3$ (MPa) | $u_0$ (MPa s$^{-1}$) | $v_0$ (MPa s$^{-1}$) | $\sigma_c$ (MPa) | $\varepsilon_c$ ($10^{-2}$) |
|---|---|---|---|---|---|---|---|
| 1 | 1-6-1 | — | 6 | 0.01 | 0.001 | 55.46 | 1.568 |
| | 1-6-2 | — | 6 | 0.1 | 0.01 | 54.63 | 1.451 |
| | 1-7-1 | — | 7 | 0.01 | 0.001 | 59.58 | 1.571 |
| | 1-7-2 | — | 7 | 0.1 | 0.01 | 58.36 | 1.462 |
| | 1-8-1 | — | 8 | 0.01 | 0.001 | 66.28 | 1.608 |
| | 1-8-2 | — | 8 | 0.1 | 0.01 | 65.56 | 1.513 |
| 2 | 2-6-1 | 3 | 6 | 0.01 | 0.001 | 52.83 | 1.502 |
| | 2-6-2 | 3 | 6 | 0.1 | 0.01 | 52.57 | 1.457 |
| | 2-7-1 | 3 | 7 | 0.01 | 0.001 | 59.00 | 1.501 |
| | 2-7-2 | 3 | 7 | 0.1 | 0.01 | 59.02 | 1.502 |
| | 2-8-1 | 3 | 8 | 0.01 | 0.001 | 65.55 | 1.520 |
| | 2-8-2 | 3 | 8 | 0.1 | 0.01 | 64.87 | 1.515 |

reaches 12 MPa, the loading and unloading speed is set to 0.1 MPa s$^{-1}$. The loading speed is set to 0.01 MPa s$^{-1}$. The rest can be deduced from this example. Initial hydrostatic stress is expressed by $\sigma_3$. The initial rate of loading axial stress is expressed in $u_0$. The initial rate $v_0$ of unloading confining stress is expressed. The strength at failure is expressed by $\sigma_c$ and the axial strain at failure is expressed by $\varepsilon_c$. The initial confining pressure was set to 6, 7 and 8 MPa, respectively. The rate of axial loading stress was set to 0.01 and 0.1 MPa s$^{-1}$. And the corresponding rate of unloading confining stress was set to 0.001 and 0.01 MPa s$^{-1}$. The test procedures are shown as follows.

Scheme 1—conventional triaxial loading–unloading test: first, an isotropic *in situ* stress state of $\sigma_1 = \sigma_2(\sigma_3)$ was applied to the specimen. Then, after a certain period of time, axial loading stress and unloading confining stress were exerted as the preset initial rate until reaching the peak stress. Finally, the axial stress-control was changed into displacement-control at a rate of 0.1 mm min$^{-1}$.

Scheme 2—alternating triaxial loading–unloading test: first, the isotropic stress was the same as scheme 1 above. After a certain period of time axial loading stress and unloading confining stress were exerted as the preset initial rate, then changing the rate of both axial loading stress and unloading confining stress rapidly and simultaneously when the axial stress was exerted by 3 MPa. For example, the rate of initial axial loading stress was 0.01 MPa s$^{-1}$ and the rate of initial unloading confining stress was 0.001 MPa s$^{-1}$; they were changed to 0.1 and 0.01 MPa s$^{-1}$ simultaneously when the axial stress was exerted by 3 MPa. After the axial stress increased 3 MPa again, the loading rate and the unloading rate were converted one more time. The steps above were repeated until reaching the peak stress. Finally, the axial stress-control was changed into displacement-control at a rate of 0.1 mm min$^{-1}$.

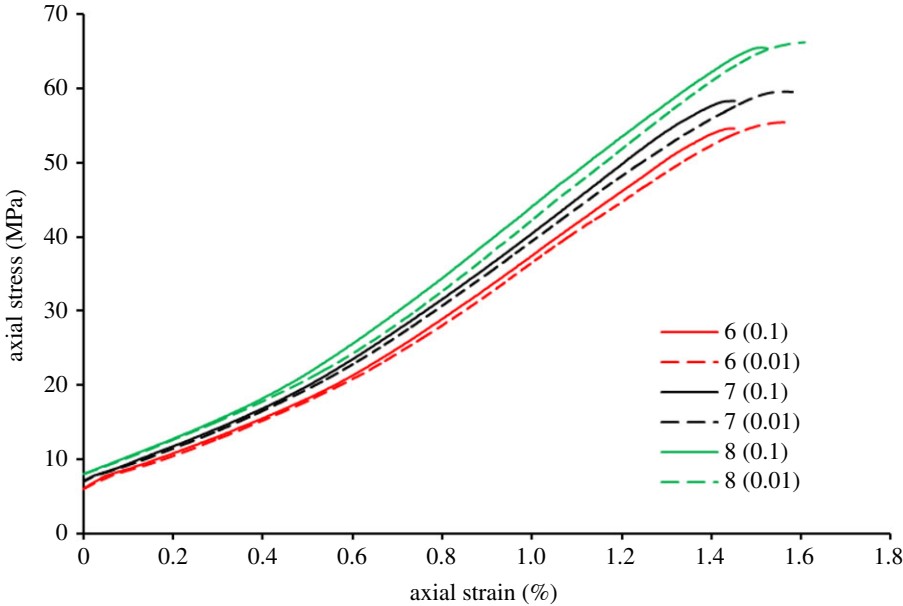

**Figure 3.** Stress–strain curves of conventional loading–unloading test before the strength of specimens.

# 3. Experimental results and analysis

## 3.1. Conventional triaxial loading–unloading test

Owing to the ratio (loading axial stress rate/unloading confining stress rate) being a fixed value, we used the loading axial stress rate to represent the rate of loading–unloading test (when the loading axial stress rate is 0.01 MPa s$^{-1}$ and the corresponding unloading confining stress rate is 0.001 MPa s$^{-1}$, replace them with the loading–unloading rate of 0.01 MPa s$^{-1}$). Figure 3 presents the stress–strain curves of conventional loading–unloading test before the peak strength. 6 (0.1) in figure 3 indicates a stress–strain curve of sample no. 1-6-2 in experiment scheme 1, wherein 6 indicates an initial hydrostatic pressure of 6 MPa, and 0.1 indicates the loading–unloading rate of 0.1 MPa s$^{-1}$. 6 (0.01) in figure 3 indicates a stress–strain curve of sample no. 1-6-1 in experiment scheme 1, wherein 6 indicates an initial hydrostatic pressure of 6 MPa, and 0.01 indicates the loading–unloading rate of 0.01 MPa s$^{-1}$. Similarly, 7 (0.1), 7 (0.01), 8 (0.1) and 8 (0.01) in figure 3 can be deduced from these examples, which are only the difference in initial hydrostatic pressure. As shown in table 1 and figure 3, when the loading–unloading rate is a fixed value, the strength and the peak axial strain of sandstone specimens increased with the increasing initial hydrostatic stress. When the loading–unloading rate was 0.01 MPa s$^{-1}$, as the initial hydrostatic stress increased from 6, 7 to 8 MPa, the strength of specimens was increased by approximately 7.43%–19.51% and the peak axial strain was increased by 0.83%–2.55%. Similarly, when the loading–unloading rate was 0.1 MPa s$^{-1}$, as the initial hydrostatic stress increased, the strength of specimens was increased by 6.83%–20.01%, and the peak axial strain was increased by 0.07%–4.27%. When the initial hydrostatic stress is a fixed value, the strength and the peak axial strain of sandstone specimens decreased with the accelerating loading–unloading rate. When the initial hydrostatic stress is 6 MPa, as the loading–unloading rate increased from 0.01 to 0.1 MPa s$^{-1}$, the strength and the peak axial strain of sandstone specimens were decreased by 1.5% and 7.46%, respectively. When the initial hydrostatic stress is 7 MPa, as the loading–unloading rate increased, the strength and the peak axial strain of sandstone specimens were decreased by 2.05% and 6.94%, respectively. When the initial hydrostatic stress is 8 MPa, as the loading–unloading rate increased, the strength and the peak axial strain of sandstone specimens were decreased by 1.07% and 5.91%, respectively.

## 3.2. Alternating triaxial loading–unloading test

Figure 4 presents the stress–strain curves of alternating loading–unloading test before the peak strength. Combining table 1 and figure 4, take 8 (0.1 → 0.01) as an example. The meaning of 8 (0.1 → 0.01)

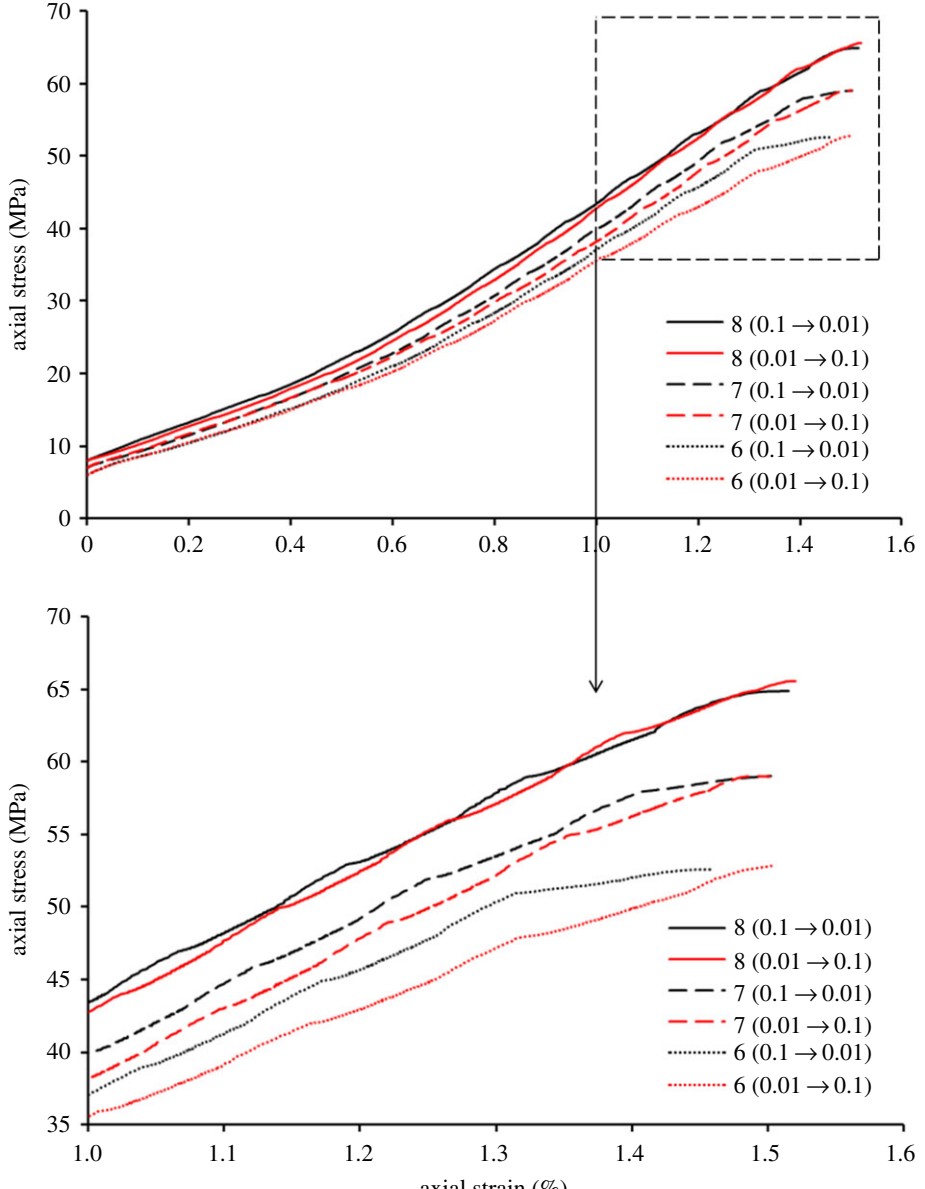

**Figure 4.** Stress–strain curves of alternating loading–unloading test before the peak strength of specimens.

in figure 4 indicates the stress–strain curve of the sample no. 2-8-2 in experimental scheme 2, 8 represents that the initial hydrostatic pressure is 8 MPa, and $(0.1 \rightarrow 0.01)$ represents that the initial loading–unloading rate of alternating triaxial loading–unloading test is 0.1 MPa s$^{-1}$. The transforming gap of converting loading–unloading rate is the time when axial stress is added by 3 MPa. For example, when the axial pressure reaches the initial hydrostatic pressure of 8 MPa, the loading and unloading speed is set to 0.1 MPa s$^{-1}$; when the axial pressure reaches 11 MPa, the loading and unloading speed is set to 0.01 MPa s$^{-1}$; and when the axial pressure reaches 14 MPa, the loading and unloading speed is set to 0.1 MPa s$^{-1}$, and continues with this change until it is destroyed, as shown in the figure. When the initial hydrostatic stress is a fixed value, the slope of stress–strain curve increased with the accelerating loading–unloading rate and decreased with the decreasing loading–unloading rate. The reason for this is that different loading–unloading rates have different effects on the mechanical properties of rock mass. When the loading–unloading rate is 0.1 MPa s$^{-1}$ (which is relatively faster), it means the pores and fissures of sandstone did not have enough time to enlarge, and the axial strain is relatively smaller in this condition. Therefore, the slope of stress–strain curve is relatively larger. When the loading–unloading rate is 0.01 MPa s$^{-1}$, compared with the rate of 0.1 MPa s$^{-1}$, the loading–unloading rate is slower, giving the pores and fissures enough time to enlarge. Therefore, the axial strain is larger, and the slope of stress–strain curve is flatter. With the development of

alternating triaxial loading–unloading test, the stress–strain curve of specimen forwards like a wave. The fluctuation of the curve first decreases then increases, and its amplitude is the largest near the peak strength. The phenomenon is caused by the evolution of the pores and fractures in sandstones. In the first transforming gap of converting rate, the original pores and fractures were the easiest to be compacted. Therefore, the change in axial strain was relatively larger. Then, as the test proceeds, the original pores and fractures were more and more difficult to be compacted and closed. The new fractures have not been formed, causing the change in axial strain to decrease. With the progress of the experiment, the new fractures generated and evolved, and the change in axial strain began to increase. The formation of the macro-fracture makes the maximum change in axial strain near the peak strength. The fluctuation of curves is different under different initial hydrostatic stress. In order to describe the performance of the fluctuation, the stability of curve-fluctuation was introduced, and it can be defined as the range of the change in axial strain under each transforming gap of rate in the whole test. The smaller the range, the higher the stability. From the results, we can know when the initial hydrostatic stress is 6 MPa, the range of the change of axial strain is around $0.061 \times 10^{-2}$–$0.137 \times 10^{-2}$. When the initial hydrostatic stress is 7 MPa, the range of the change of axial strain is around $0.058 \times 10^{-2}$–$0.122 \times 10^{-2}$; When the initial hydrostatic stress is 8 MPa, the range of the change of axial strain is around $0.053 \times 10^{-2}$–$0.107 \times 10^{-2}$. These show that the stability of curve-fluctuation increased with the increasing initial hydrostatic stress because the effect of different restraint stress on axial strain is different. When the initial hydrostatic stress is low, the specimen is equivalent to uniaxial compression state before failure. Along with the experimental proceeding, the confining pressure stress becomes lower and lower, and the confinement effect of confining pressure on axial strain is getting smaller and smaller. The change in axial strain caused by loading and unloading is getting larger and larger in the stage of developing fractures. Therefore, the stability of curve-fluctuation is poor. However, with the increase of initial hydrostatic stress, the confining stress before the peak strength is relatively higher. Under this condition, the confinement effect of confining pressure on axial strain tends to be the same as that in the whole process. Therefore, the change in axial stress in each transforming gap of rate tends to be equal, and the fluctuation of curves tends to be stable in the whole alternating triaxial loading–unloading tests.

Different initial speeds resulted in different coincidences degree of stress–strain curves in alternating triaxial loading–unloading tests. The reason leading to this phenomenon is that different loading and unloading speeds have different effects on axial strain under different confining pressures for the same speed change interval. When the confining stress was higher, the change of axial strain with high rate was closer to the change of axial stain with low rate under the same transforming gap of the rate. Therefore, the coincidence of stress–strain curves was higher. On the contrary, with the decrease of confining stress, the variation in axial strain under different loading–unloading rates increased, and the deviation of stress–strain curve increased slowly. When the initial hydrostatic stress of test was higher, the confining stress before failure was relatively higher. In the last transforming gap of rate, the effect of confining stress on axial strain is still large, and the change in axial strain under different loading–unloading rates remains similar. Therefore, in the whole alternating triaxial loading–unloading test, the coincidence degree of stress–strain curves under different initial rates remained higher significantly. With the decrease of initial hydrostatic stress, the confining stress in each transforming gap of rate reduces relatively, leading to the diversity of axial strain increased, and the deviation of stress–strain curves increased in the whole test. Therefore, the coincidence of stress–strain curves with different initial rates increased with the increasing initial hydrostatic stress in alternating triaxial loading–unloading tests.

As shown in figure 5, the stress–strain curves of conventional triaxial loading–unloading test and alternating triaxial loading–unloading test have been compared. The results show that when the initial rate of alternating triaxial loading–unloading test is 0.1 MPa s$^{-1}$, the turning points (the loading–unloading rate converted from 0.1 to 0.01 MPa s$^{-1}$) of stress–strain curve coincide well within the stress–strain curves of conventional triaxial loading–unloading tests (with the initial rate is 0.1 MPa s$^{-1}$), under the condition of the initial hydrostatic stresses of 7 and 8 MPa. When the initial hydrostatic pressure is 6 MPa, except for the difference near the peak strength, the other places can coincide well, showing good experimental results. With the increase of initial hydrostatic pressure, the coincidence degree between turning point A (loading and unloading speed from 0.01 to 0.1 MPa s$^{-1}$ in stress–strain curve) and turning point B (conventional loading and unloading speed of 0.01 MPa s$^{-1}$ in stress–strain curve) increases. The 3 MPa transforming gap of rate and sandstone samples with high strength, these factors determine the limited change of axial strain and the little fluctuation of alternating stress–strain curve. With the increase of initial hydrostatic stress, the corresponding confining stress in the test also increased, and the change of axial strain under

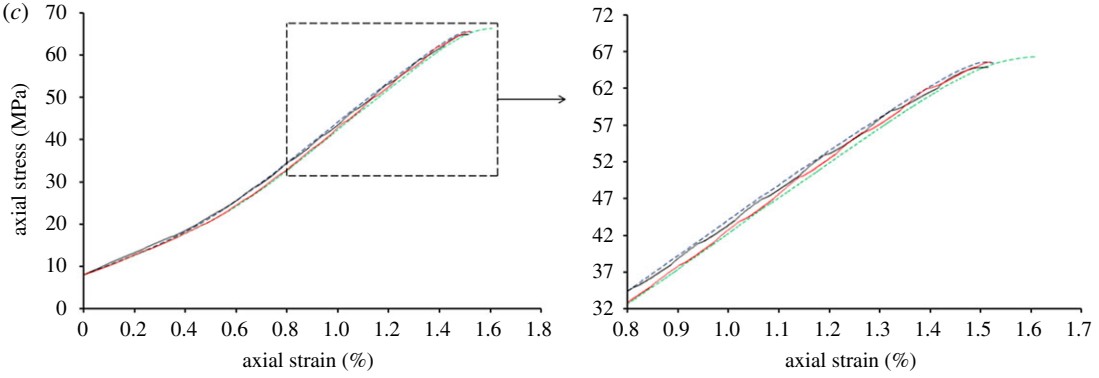

**Figure 5.** Comparison chart of stress–strain curves between alternating loading–unloading test and conventional loading–unloading test. The initial hydrostatic stress is (*a*) 6 MPa, (*b*) 7 MPa and (*c*) 8 MPa.

different loading–unloading rates was similar to the condition of adding the same axial stress. Therefore, the stress–strain curves of conventional loading–unloading tests with different initial rate were closer to each other, coinciding well with the turning points of alternating triaxial loading–unloading test. When the initial speed of alternating loading and unloading is 0.01 MPa s$^{-1}$, the coincidence between the turning point of the stress–strain curve and the conventional loading and unloading experimental curve increases with the increase of the initial hydrostatic pressure, but its overall coincidence effect is worse than that of the alternating loading and unloading experiment with the initial speed of 0.1 MPa s$^{-1}$.

On the stress–strain curves of alternating loading and unloading specimens, the turning point of loading and unloading speed from 0.1 to 0.01 MPa s$^{-1}$ is used to fit the curve. The fitting curve simulates the stress–strain curve of loading and unloading speed of 0.1 MPa s$^{-1}$. In the same way, fitting the turning points by changing the loading–unloading rate from 0.01 to 0.1 MPa s$^{-1}$ simulates the stress–strain curve of conventional triaxial loading–unloading test with a rate of 0.01 MPa s$^{-1}$. Drawing the fitting curves and the stress–strain curves of conventional triaxial loading–unloading test in the same coordinate system, the distance between two curves was defined as similarity. The smaller the distance, the larger the similarity. Comparing the similarity between the fitted curve and the conventional loading and unloading curve, it can be found that the similarity increases with the

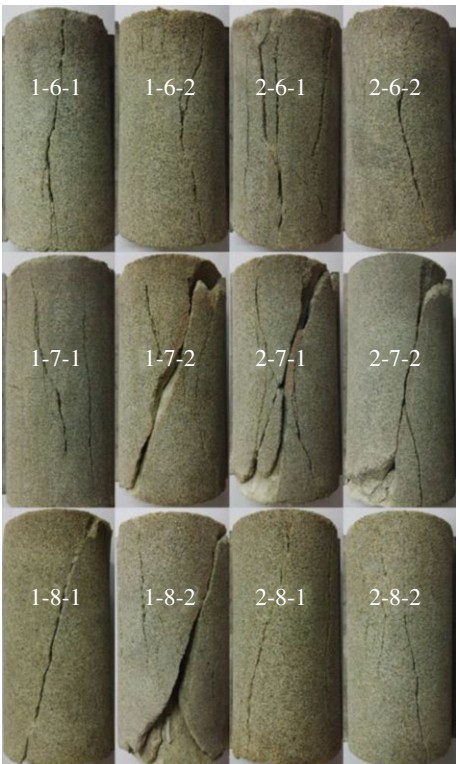

**Figure 6.** Destruction form of sandstone specimens.

increase of the initial hydrostatic pressure, and increases with the increase of the initial acceleration and unloading speed. Therefore, if the inflection point corresponding to the stress–strain curve obtained by alternate loading and unloading is fitted, the higher the initial hydrostatic pressure of the fitting curve, the more realistic simulation of the stress–strain curve for conventional loading and unloading experiments. From the previous research, we can know that the change of mining-induced stress caused by excavation could be simulated by the conventional loading–unloading test. With the increase of excavation depth, the stress of rock is getting larger. In the laboratory simulation experiment, it is shown as the conventional loading–unloading test under high initial hydrostatic stress. Combined with the research of the experiment, the mechanical properties of deep rock under different excavation rates could been studied by alternating triaxial loading–unloading test. The experiment provides better results when the initial rate of alternating triaxial loading–unloading test is relatively higher. Through this method, we can obtain the stress–strain curves of different loading–unloading rate on a single specimen by fitting the stress–strain curves of alternating triaxial loading–unloading test. Then, the effect of the loading–unloading rate on the mechanical properties of rock mass could be analysed, in the case of eliminating the dispersion of different specimens. Finally, the results obtained this way would be more accurate.

## 4. Destruction form

Figure 6 presents the destruction form of sandstone specimens. The specimens destroyed with a clear and crisp 'bang' sound during the experiment, representing the typical properties of brittle failure. The specimens destroyed with micro-drum had obvious lateral expansion. Along with the increasing initial hydrostatic stress, the failure behaviour of the sandstone samples tested under loading–unloading methods changed from a tensile state to a tensile-shear coexisting state, and finally to a fully shear failure state. In the condition of low confining stress, the macro-fractures extended to both sides of the specimen. All kinds of tension fractures and micro-fractures extended fully, and the inclination angle of the main fracture was almost 80–90°, nearly parallel to the major principal stress ($\sigma_1$). The tension fractures and the inclination angle of the main fracture reduced with the increase of confining stress. The reason is that when the loading axial stress is equal to the unloading confining stress, the specimen will change from triaxial compression to uniaxial compression. When the initial hydrostatic

stress is at a lower value, the confining stress before the peak strength is lower, which is equivalent to the uniaxial compression state, and the main failure mode of the specimen is in tensional failure mode. With the increase of initial hydrostatic stress, the confining stress before the peak strength also increased, and the destruction form of specimen changed into shear state. Furthermore, the degree of destruction increased with the accelerating rate of loading–unloading under the same initial hydrostatic stress.

## 5. Conclusion

The study designs a new loading–unloading experimental method for simulating the change of mining-induced stress under two testing schemes. The stress–strain curve before the peak strength in conventional and alternating triaxial loading–unloading test were studied and compared. The similarity between fitting curves and conventional stress–strain curves under different conditions were explored. Furthermore, the destruction rule of specimens was discussed. Based on these results, several understandings and conclusions can be drawn.

First, when the loading–unloading rate is a fixed value, the strength and the peak axial strain of sandstone specimens increased with the increasing initial hydrostatic stress. When the initial hydrostatic stress is a fixed value, the strength and the peak axial strain of sandstone specimens decreased with the accelerating loading–unloading rate.

Second, with the proceeding of alternating triaxial loading–unloading test, stress–strain curve of specimen forwards like a wave. The fluctuation of the curve first decreases then increases and achieves the maximum near the peak strength. Its stability increased with the increasing initial hydrostatic stress.

Third, the similarity between the fitting stress–strain curves obtained by alternating triaxial loading–unloading test and the stress–strain curves by conventional triaxial loading–unloading tests will be affected by the initial hydrostatic pressure and the initial rate of the alternating loading–unloading test, the similarity increased with the increasing initial hydrostatic stress, and increased with the accelerating initial rate of alternating loading–unloading test. The fitted curve can be formed by fitting the corresponding inflection point in the stress–strain curve of the alternating triaxial loading–unloading tests, the fitting curve could represent the stress–strain curve of conventional triaxial loading–unloading test more accurately under the higher initial hydrostatic stress.

Fourth, along with the increasing initial hydrostatic stress, the failure behaviour of the sandstone samples tested under loading–unloading methods changed from a tensile state to a tensile-shear coexisting state, and finally to a fully shear failure state. Furthermore, the degree of destruction increased with the accelerating loading–unloading rate under the same initial hydrostatic stress.

Fifth, the mechanical properties of deep rock under different excavation rates could been studied by alternating triaxial loading–unloading test. The experiment provides better results when the initial rate of alternating triaxial loading–unloading test is relatively higher.

Permission to carry out fieldwork. Permission to collect sandstone was from the local government in Chongqing City, China.
Data accessibility. Data have been uploaded as electronic supplementary material.
Authors' contributions. X.L. and H.Y. carried out the molecular laboratory work, participated in data analysis, carried out sequence alignments, participated in the design of the study and drafted the manuscript; X.L. and M.L. carried out the statistical analyses; H.Y. collected field data; C.J. and M.D. conceived of the study, designed the study, coordinated the study and helped draft the manuscript. All authors gave final approval for publication.
Competing interests. We have no competing interests.
Funding. This study was financially supported by the National Natural Science Foundation of China (51674048) and Fundamental and Advanced Research Projects of Chongqing (cstc2015jcyjA90009).
Acknowledgements. We thank Binwei Niu, Tianyu Lu and Jun Lin for their support of the experiment of our study. We are also grateful to Tang Yu, Aidong Wei and Zhuo Jin for their help in analysing the data.

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
