## [Reviewer comments · Royal Society Open Science]

Review History

RSOS-181964.R0 (Original submission)

Review form: Reviewer 1

Is the manuscript scientifically sound in its present form?

Yes

Are the interpretations and conclusions justified by the results?

Yes

Is the language acceptable?

Yes

Is it clear how to access all supporting data?

Yes

Do you have any ethical concerns with this paper?

No

Have you any concerns about statistical analyses in this paper?

No

Recommendation?

Accept with minor revision (please list in comments)

Comments to the Author(s)

The manuscript carried out alternating loading and unloading experiments under different confining pressures and different loading-unloading speeds, alternating the rate of loading-unloading on the sandstone sample. A new experimental method for obtaining different experimental curves of loading-unloading speed on the one sample was creatively studied in the manuscript, and the feasibility of the new method was verified. The experimental method has certain innovation and practicality, and provides a new idea to eliminate sample dispersion.

However, there are still some things that need minor improvements.

1. It is recommended to provide more detailed information or physical parameters (such as location, density.) of the sample so that other researchers can judge and understand the work of the paper.
2. It is best to explain in detail the meaning of $\Delta \sigma$ on the 21st line of page 4, and the meanings of the lower right corner of Figure 3 and Figure 4 (such as 6(0.1), 6(0.01)).
3. English should be checked again, there are some spelling mistakes in the paper, such as the semicolon in the abstract, the words after the comma should be lowercase, I suggest a native English speaker check it English before re-submission.

Review form: Reviewer 2

Is the manuscript scientifically sound in its present form?

Yes

Are the interpretations and conclusions justified by the results?

Yes

Is the language acceptable?

Yes

Is it clear how to access all supporting data?

Yes

Do you have any ethical concerns with this paper?

No

Have you any concerns about statistical analyses in this paper?

No

Recommendation?

Accept with minor revision (please list in comments)

Comments to the Author(s)

I wrote some suggestions to authors of this article (Appendix A).

Decision letter (RSOS-181964.R0)

08-Apr-2019

Dear Professor Jiang

On behalf of the Editors, I am pleased to inform you that your Manuscript RSOS-181964 entitled "A new loading-unloading experimental method for simulating the change of mining-induced stress" has been accepted for publication in Royal Society Open Science subject to minor revision in accordance with the referee suggestions. Please find the referees' comments at the end of this email.

The reviewers and handling editors have recommended publication, but also suggest some minor revisions to your manuscript. Therefore, I invite you to respond to the comments and revise your manuscript.

- Ethics statement

- Data accessibility

If you wish to submit your supporting data or code to Dryad (<http://datadryad.org/>), or modify your current submission to dryad, please use the following link:
<http://datadryad.org/submit?journalID=RSOS&manu=RSOS-181964>

- Competing interests

- Authors' contributions

- Acknowledgements

- Funding statement

Because the schedule for publication is very tight, it is a condition of publication that you submit the revised version of your manuscript before 17-Apr-2019. Please note that the revision deadline will expire at 00.00am on this date. If you do not think you will be able to meet this date please let me know immediately.

- 1) A text file of the manuscript (tex, txt, rtf, docx or doc), references, tables (including captions) and figure captions. Do not upload a PDF as your "Main Document";
- 2) A separate electronic file of each figure (EPS or print-quality PDF preferred (either format should be produced directly from original creation package), or original software format);
- 3) Included a 100 word media summary of your paper when requested at submission. Please ensure you have entered correct contact details (email, institution and telephone) in your user account;
- 4) Included the raw data to support the claims made in your paper. You can either include your data as electronic supplementary material or upload to a repository and include the relevant doi

within your manuscript. Make sure it is clear in your data accessibility statement how the data can be accessed;

5) All supplementary materials accompanying an accepted article will be treated as in their final form. Note that the Royal Society will neither edit nor typeset supplementary material and it will be hosted as provided. Please ensure that the supplementary material includes the paper details where possible (authors, article title, journal name).

on behalf of Professor Jon Blundy (Subject Editor)
openscience@royalsociety.org

Reviewer comments to Author:

Reviewer: 1

Comments to the Author(s)

The manuscript carried out alternating loading and unloading experiments under different confining pressures and different loading-unloading speeds, alternating the rate of loading-unloading on the sandstone sample. A new experimental method for obtaining different experimental curves of loading-unloading speed on the one sample was creatively studied in the manuscript, and the feasibility of the new method was verified. The experimental method has certain innovation and practicality, and provides a new idea to eliminate sample dispersion.

However, there are still some things that need minor improvements.

1. It is recommended to provide more detailed information or physical parameters (such as location, density.) of the sample so that other researchers can judge and understand the work of the paper.

2. It is best to explain in detail the meaning of $\Delta \sigma$ on the 21st line of page 4, and the meanings of the lower right corner of Figure 3 and Figure 4 (such as 6(0.1), 6(0.01)).
3. English should be checked again, there are some spelling mistakes in the paper, such as the semicolon in the abstract, the words after the comma should be lowercase, I suggest a native English speaker check it English before re-submission.

Reviewer: 2

Comments to the Author(s)

I wrote some suggestions to authors of this article (in attachment)

Author's Response to Decision Letter for (RSOS-181964.R0)

See Appendix B.

Decision letter (RSOS-181964.R1)

21-May-2019

Dear Professor Jiang,

I am pleased to inform you that your manuscript entitled "A new loading-unloading experimental method for simulating the change of mining-induced stress" is now accepted for publication in Royal Society Open Science.

on behalf of Mr Andrew Dunn (Associate Editor) and Jon Blundy (Subject Editor)
openscience@royalsociety.org

Associate Editor Comments to Author (Mr Andrew Dunn):

Associate Editor: 1

Comments to the Author:

(There are no comments.)

Reviewer comments to Author:

Appendix A

Comments:

1. The article may be interesting from the point of view of mining and also during any construction works requiring e.g. tunnel drilling. Considering the practical aspect of the research, reading the article, I noticed the lack:
 - Direct reference to mining. The part of the title of the article is "simulating the change of mining-induced stress". The authors focused on presenting the result of the experiment, but without presenting what these research actually bring to mining practice. Do they have any practical applications? If so, it has not been clearly articulated. The title suggests that the work is related to mining, but it does not really follow from the content.
 - I did not see a clear purpose of research. If the authors consider the text from page 3 19-21 as a purpose of research ("Through the method of the test, the mechanical characteristics of characteristics loading characteristics loading-unloading rates were more accurately, and the theoretical guidance of rock excavation was more scientifically"), they maybe should make reconstruction of the title or also purpose of research. The contents of the article suggest more theoretical and laboratory practical considerations rather than the impact on mining...
2. I understand that the authors of the article for the purpose of the research use a device constructed for the analysis of coal containing methane (Triaxial servo-controlled seepage equipment for thermo-hydro-mechanical coupling of coal containing methane). The article in which they describe the device (In references No. 29 - It was printed in the Chinese magazine) is not generally available, so I suggest to include even a brief description of the device (this is self-made so this is interesting for readers). If this device was made for coal research, why you decided to apply it to sandstone samples?
3. Sampling - why was this rock chosen? Is a device made for coal suitable for a different rock? The name has clearly to "coal". Do the authors know the characteristics of the analysed sandstone. Can you assume this sandstone is fairly homogeneous, as isotropic? Because it ensures the repeatability of the measurement. The authors write (p.3, 57-58) about "without visible fractures and cracks by means of photo observation". Which means "photo observation"? Did you watch it under any microscopic magnification? Did you use any micro- or only macro analysis ? Were any petrographic analyses carried out? Is composition is known (skeleton, matrix, cement)? Do you know any physical properties of these rocks?
4. The article contains a fairly wide references review, but it focuses on samples other than sandstones - the authors write about mechanical properties: coal, marble salt rocks, granite, migmatite, and even man-made rocks (similar as sandstones but more homogeneous) e.t.c. But the part about the behaviour of sandstones and research on them is relatively poorly developed. Sandstones are, after all, very common rocks, especially as rocks accompanying in mines. Sandstone research has been and is being carried out on a large scale, which is why I think it is worth reaching deeper into the literature on the mechanical properties of sandstone. There are a number of works dealing with clastic rocks in the context of strength.

Conclusion:

The article can be printed, but I suggest a deeper analysis of the sandstone references, maybe rewording the title or purpose of the research, as well as a brief description of the device and method,

explaining why this device was applied to the sandstone. And in the future, a little more emphasis on the physical and petrographic properties of used samples.

Appendix B

Response to Editor and Reviewers

Dear Editor and Reviewers,

We deeply appreciate the critical comments concerning our manuscript. These comments are very valuable and helpful for improving our paper. We have studied all the comments carefully and have made corrections accordingly.

The main corrections in the paper and the responses to the reviewers' comments are given as follows.

Revisions in the text are shown using **red highlight** for additions, and **blue highlight** for grammatical checks. We hope that the revisions in the manuscript and our accompanying responses will be sufficient to make our manuscript suitable for publication in *Royal Society Open Science*.

We deeply appreciate your critical comments concerning our manuscript. The manuscript has been resubmitted to the journal and we look forward to your positive response at your earliest convenience.

Very truly yours,
Changbao Jiang

Corresponding author: Changbao Jiang
State Key Laboratory of Coal Mine Disaster Dynamics and Control
Chongqing University, Chongqing 400030
China
Tel.: +86 23 65111228
Fax: +86 23 65111468
E-mail address: jcb@cqu.edu.cn

Reviewer 1

Issue 1: *It is recommended to provide more detailed information or physical parameters (such as location, density.) of the sample so that other researchers can judge and understand the work of the paper.*

Discussion: Revision at Page 5 Lines 7-Page 5 Lines 14

The sandstone samples of this experiment were taken from the roof of a coal mine in Chongqing, China. The apparent density ranged from 2237 to 2243 kg/m³. The mineral composition mainly includes feldspar and quartz. The content of quartz is less than 75%, and the content of feldspar is more than 25 %, and the amount of cuttings is less than 25% [42]. The sandstone has a moisture content of 0.44%, an ash content of 96.21%, a volatile content of 3.26%, a fixed carbon of 0.09%, a TOC content of 0.1%, and a porosity of 4.15%. The sandstone has good homogeneity, small dispersion and good sampling integrity. The experimental samples were taken from the same intact large sandstone.

Issue 2: *It is best to explain in detail the meaning of $\Delta\sigma$ on the 21st line of page 4, and the meanings of the lower right corner of Figure 3 and Figure 4 (such as 6(0.1), 6(0.01)).*

Discussion: Revision at Page 5 Lines 19-Page 5 Lines 24, Page 6 Lines 21-Page 6 Lines 27 and Page 7 Lines 11-Page 7 Lines 19.

$\Delta\sigma$: In the second experimental scheme, after the axial pressure is increased by a certain value, the loading and unloading speed is simultaneously changed. $\Delta\sigma$ represents the speed alternately changing interval with the increase in the axial pressure value, that is, the load-unloading speed is simultaneously changed every time the axial pressure is increased by a certain stress value (3 MPa). For example, the experimental scheme of sample 2-6-1 in Table 1 is that when the axial pressure is 6 MPa of the initial hydrostatic pressure, the load-unloading speed is 0.01 MPa/s, and when the axial pressure is 9 MPa, the transformation load-unloading speed is 0.1 MPa/ s, when the axial pressure is 12 MPa, the transformation load-unloading speed is 0.01 MPa/s, then can deduce the rest from this example.

The meanings of the lower right corner of Figure 3 and Figure 4:

6 (0.1) in Fig. 3 indicates a stress-strain curve of sample No. 1-6-2 in experiment scheme 1, wherein 6 indicates an initial hydrostatic pressure of 6 MPa, and 0.1 indicates the loading-unloading rate of 0.1 MPa/s. 6 (0.01) in Fig. 3 indicates a stress-strain curve of sample No. 1-6-1 in experiment scheme 1, wherein 6 indicates an initial hydrostatic pressure of 6 MPa, and 0.01 indicates the loading-unloading rate of 0.01 MPa/s. Similarly, 7 (0.1), 7 (0.01), 8 (0.1), 8 (0.01) in Fig. 3 can deduce the rest from these examples, which are only the difference in initial hydrostatic pressure.

6 (0.01 → 0.1) in Fig. 4 indicates the stress-strain curve of sample No. 2-6-1 in the experimental scheme 2, wherein 6 indicates that the initial hydrostatic pressure is 6 MPa, and (0.01 → 0.1) indicates that the initial loading-unloading rate is 0.01MPa/s, the conversion loading-unloading speed interval is the time when the axial stress increases by 3 MPa. For example, when the axial pressure is the initial hydrostatic pressure of 6 MPa, the loading-unloading speed is 0.01MPa/s, and when the axial

pressure is 9 MPa, the axial compression is changed. The loading-unloading speed is 0.1 MPa/s, and the loading-unloading speed is 0.01 MPa/s when the shaft is pressed to 12 MPa, and goes on with this change until it breaks down. 6 (0.1→0.01) in Fig. 4 indicates the stress-strain curve of the sample No. 2-6-2 in the experimental scheme 2, wherein 6 indicates that the initial hydrostatic pressure is 6 MPa, and (0.1→0.01) indicates that the initial loading-unloading speed is 0.1 MPa/s, the conversion loading-unloading speed is the time when the axial stress increases by 3MPa. For example, when the axial pressure is the initial hydrostatic pressure of 6MPa, the loading-unloading speed is 0.1 MPa/s, and when the axial pressure is 9MPa, the transformation loading-unloading speed is 0.01 MPa/s, when the axial pressure is 12 MPa, the transformation loading-unloading speed is 0.1 MPa/s, and continues with this change until it is destroyed. Similarly, 7 (0.01 → 0.1), 7 (0.1 → 0.01), 8 (0.01 → 0.1), 8 (0.1 → 0.01) in Fig. 4 can deduce the rest from above examples, which are only the difference of the initial hydrostatic pressure.

Issue 3: English should be checked again, there are some spelling mistakes in the paper, such as the semicolon in the abstract, the words after the comma should be lowercase, I suggest a native English speaker check it English before re-submission.

Discussion:

Thanks to the reviewer's suggestion, we sought the help of the translation company to polish the English, and the specific changes were marked in blue highlight.

Reviewer 2

Issue 1: The article may be interesting from the point of view of mining and also during any construction works requiring e.g. tunnel drilling. Considering the practical aspect of the research, reading the article, I noticed the lack:

- Direct reference to mining. The part of the title of the article is "simulating the change of mining-induced stress". The authors focused on presenting the result of the experiment, but without presenting what these research actually bring to mining practice. Do they have any practical applications? If so, it has not been clearly articulated. The title suggests that the work is related to mining, but it does not really follow from the content.

- I did not see a clear purpose of research. If the authors consider the text from page 3 19-21 as a purpose of research ("Through the method of the test, the mechanical characteristics of characteristics loading characteristics loading-unloading rates were more accurately, and the theoretical guidance of rock excavation was more scientifically"), they maybe should make reconstruction of the title or also purpose of research. The contents of the article suggest more theoretical and laboratory practical considerations rather than the impact on mining...

Discussion: Revision at Page 4 Lines 11- Page 4 Lines 14

Many thanks to the reviewer for your advice. Xie (academician of the Chinese Academy of Engineering) et al. [39] proposed that the rock mass experienced a complete kinetic process from the original rock stress, axial stress rise (loading) and confining pressure reduction (unloading) to failure unloading when mining underground in front of the working face. The variation of mining stress can be studied by stress path experiments of rock under loading axial pressure and unloading confining pressure. The surrounding rock in the mining process is often sedimentary rock, such as sandstone, mudstone, limestone and other rocks. Therefore, we have proposed a new type of loading and unloading experiment to simulate the change of mining stress. The next step of our research is to analyze the correspondence between the mining stress in the laboratory simulation and the mining stress change caused by mining in the actual engineering problems.

Issue 2: I understand that the authors of the article for the purpose of the research use a device constructed for the analysis of coal containing methane (Triaxial servo-controlled seepage equipment for thermo-hydro-mechanical coupling of coal containing methane). The article in which they describe the device (In references No. 29 - It was printed in the Chinese magazine) is not generally available, so I suggest to include even a brief description of the device (this is self-made so this is interesting for readers). If this device was made for coal research, why you decided to apply it to sandstone samples?

Discussion: Revision at Page 4 Lines 20- Page 4 Lines 26

The revision modifies the English reference for the instrument (It turned out to be [29], now it is [40]) and introduces the instrument briefly. The maximum axial pressure of the device can reach 100 MPa. The maximum confining pressure can reach 10 MPa. The maximum gas pressure can be up to 2 MPa and the test temperature can be up to

90 °C. Axial loads enable force control and displacement control. Parameters such as stress, deformation, gas pressure, temperature and flow can be collected automatically. The device can be applied to coal and other rock experiments, and can carry out mechanical and seepage experiments under different in-situ stress fields (confining pressure and axial pressure), different pressures and different geothermal fields. Therefore, the sandstone samples were tested using this device. Although the experimental instrument was developed for coal research, in fact the instrument can be applied to all rocks (Zhang D. M. [25] carried out sandstone experiments with this instrument). Therefore, the instrument was applied to this sandstone experiment.

Issue 3: Sampling - why was this rock chosen? Is a device made for coal suitable for a different rock? The name has clearly to "coal". Do the authors know the characteristics of the analysed sandstone. Can you assume this sandstone is fairly homogeneous, as isotropic? Because it ensures the repeatability of the measurement. The authors write (p.3, 57-58) about "without visible fractures and cracks by means of photo observation". Which means "photo observation"? Did you watch it under any microscopic magnification? Did you use any micro- or only macro analysis? Were any petrographic analyses carried out? Is composition is known (skeleton, matrix, cement)? Do you know any physical properties of these rocks?

Discussion: Revision at Page 5 Lines 7- Page 5 Lines 14

The surrounding rock in the mining process is often sedimentary rock, such as sandstone, mudstone, limestone and other rocks. The main purpose of this manuscript is to explore new methods for performing different loading and unloading rates on the same rock. In order to verify whether this method is effective, it is more ideal to use relatively homogeneous rock, and the homogeneity of sandstone is better. Therefore the manuscript uses sandstone as an experimental sample.

The sandstone samples of this experiment were taken from the roof of a coal mine in Chongqing, China. The apparent density ranged from 2237 to 2243 kg/m³. The mineral composition mainly includes feldspar and quartz. The content of quartz is less than 75%, and the content of feldspar is more than 25 %, and the amount of cuttings is less than 25% [42]. The sandstone has a moisture content of 0.44%, an ash content of 96.21%, a volatile content of 3.26%, a fixed carbon of 0.09%, a TOC content of 0.1%, and a porosity of 4.15%. The sandstone has good homogeneity, small dispersion and good sampling integrity. The experimental samples were taken from the same intact large sandstone.

Unfortunately, our laboratory lacks precision instruments such as scanning electron microscopes. Therefore, microscopic analysis experiments are not performed on the samples. Only the photos are viewed. Next, we will seek means to carry out relevant microscopic analysis experiments.

Issue 4: The article contains a fairly wide references review, but it focuses on samples other than sandstones - the authors write about mechanical properties: coal, marble salt rocks, granite, migmatite, and even man-made rocks (similar as sandstones but more homogeneous) e.t.c. But the part about the behaviour of

sandstones and research on them is relatively poorly developed. Sandstones are, after all, very common rocks, especially as rocks accompanying in mines. Sandstone research has been and is being carried out on a large scale, which is why I think it is worth reaching deeper into the literature on the mechanical properties of sandstone. There are a number of works dealing with clastic rocks in the context of strength.

Discussion: Revision at Page 2 Lines 34- Page 3 Lines 32

Less reference to the sandstone literature is my negligence, thanks to the reviewer's reminders and suggestions. After a detailed review of the literature, I added several articles on sandstone loading and unloading studies as references, ranging from [22] to [31]